# Quantitative Raman Analysis of Carotenoid Protein Complexes in Aqueous Solution

**DOI:** 10.3390/molecules27154724

**Published:** 2022-07-24

**Authors:** Joy Udensi, Ekaterina Loskutova, James Loughman, Hugh J. Byrne

**Affiliations:** 1FOCAS Research Institute, Technological University Dublin, City Campus, Camden Row, Dublin 8, D08 CKP1 Dublin, Ireland; hugh.byrne@tudublin.ie; 2School of Physics and Clinical and Optometric Sciences, Technological University Dublin, City Campus, Grangegorman, Dublin 7, D07 EWV4 Dublin, Ireland; kate.loskutova@tudublin.ie (E.L.); james.loughman@tudublin.ie (J.L.); 3Centre for Eye Research, Ireland, Technological University Dublin, City Campus, Grangegorman, Dublin 7, D07 EWV4 Dublin, Ireland

**Keywords:** carotenoids, Beta Carotene, Lutein, Zeaxanthin, Raman spectroscopy, bovine serum albumin

## Abstract

Carotenoids are naturally abundant, fat-soluble pigmented compounds with dietary, antioxidant and vision protection advantages. The dietary carotenoids, Beta Carotene, Lutein, and Zeaxanthin, complexed with in bovine serum albumin (BSA) in aqueous solution, were explored using Raman spectroscopy to differentiate and quantify their spectral signatures. UV visible absorption spectroscopy was employed to confirm the linearity of responses over the concentration range employed (0.05–1 mg/mL) and, of the 4 Raman source wavelengths (785 nm, 660 nm, 532 nm, 473 nm), 532 nm was chosen to provide the optimal response. After preprocessing to remove water and BSA contributions, and correct for self-absorption, a partial least squares model with R^2^ of 0.9995, resulted in an accuracy of the Root Mean Squared Error of Prediction for Beta Carotene of 0.0032 mg/mL and Limit of Detection 0.0106 mg/mL. Principal Components Analysis clearly differentiated solutions of the three carotenoids, based primarily on small shifts of the main peak at ~1520 cm^−1^. Least squares fitting analysis of the spectra of admixtures of the carotenoid:protein complexes showed reasonable correlation between norminal% and fitted%, yielding 100% contribution when fitted with individual carotenoid complexes and variable contributions with multiple ratios of admixtures. The results indicate the technique can potentially be used to quantify the carotenoid content of human serum and to identify their differential contributions for application in clinical analysis.

## 1. Introduction

Carotenoids represent a wide range of fat-soluble pigmented compounds found abundantly in nature [1]. Carotenoids are abundant throughout the ecosystem, found in a wide variety of plants, animals and even micro-organisms [2]. The human body is no exception, as carotenoids are an integral part of most diets, especially as they are contained in a broad range of coloured fruits and leafy vegetables [3,4]. When these foods are broken down, some carotenoids find their way into the blood stream and hence can be detected in serum using sensitive tests such as high-pressure liquid chromatography (HPLC) or optical spectroscopy [5]. They are also found in adipose tissue, liver, and muscles [6,7].

Carotenoids have received significant attention in recent years because of their reported antioxidant activity in the body [3,4,8] and research has, over time, pointed to the fact that these compounds can play an important role in protection against cancer and cardiovascular diseases [8]. They have also been shown to protect against eye diseases, such as cataracts [9,10] and age-related macular degeneration (AMD) [11,12]. Some carotenoids can be converted to retinol, which is used for detection of light in the eye [13]. Three distinct carotenoids, Lutein, Zeaxanthin, and Meso-Zeaxanthin, are found in the macula of the retina in the eye [14]. These carotenoids can absorb short wavelength light, and hence provide photic and antioxidant protection to the macula in the retina [4,15]. In a broader context, carotenoids have attracted a lot of attention because of their roles in photochemistry, photobiology, and photo medicine [3]. Carotenoids absorb light in the visible range and their unique function and properties are largely due to the existence of a long chromophore of conjugated double bonds that are the origin of their light-absorbing properties and make them very susceptible to oxidative degradation [16].

Raman spectroscopy has been explored as an alternative to the more common, but highly laborious and time-consuming, HPLC methods to detect carotenoids in fruits, vegetables, and body tissues [6]. Raman has very important advantages in that it is label/dye free, more rapid, less complicated, and highly sensitive when compared with the traditional HPLC methods [17]. During Raman spectroscopy, the strong absorption bands of carotenoids can provide resonant enhancement for the sensitive detection of its specific Raman response [5]. Because of this high sensitivity, there is now a growing body of research that advocates the use of Raman spectroscopy in detection of macular pigment carotenoids for the early diagnosis of eye diseases such as AMD and glaucoma [18]. Its antioxidant properties could also mean Raman spectroscopy can be channeled to biomarker research for detecting risk factors for cancer in individuals at a much earlier stage [19].

Various Raman studies have been carried out which show the spectroscopic responses of key carotenoids, both in their natural states [8,20] and in human tissues [7,13,15,21]. In the Raman spectrum of carotenoids, three strong signals, which come from the carbon-carbon double bonds and single bond stretches of the polyene backbone and methyl groups are usually prominent. The most prominent of these peaks can be observed at ~1519 cm^−1^, ~1158 cm^−1^, and ~1004 cm^−1^ in the carotenoid spectrum [5], and this is particularly so when the Raman source wavelength is resonant or near-resonant with the optical absorption bands of the compounds (e.g., 532 nm) [15,22]. This can become useful in the early detection of cancers [21,22], and Raman spectroscopic analysis of the macular carotenoids Lutein and Zeaxanthin has been explored to advance diagnostic studies of diseases such as AMD [22].

Beta Carotene, Lutein, and Zeaxanthin are among the most studied carotenoids. Beta Carotene, being the most abundant in the human diet, blood, and tissues, belongs to the class of carotenoids called carotenes, which contain only carbon and hydrogen atoms [23,24] Lutein and Zeaxanthin, on the other hand, are members of a class of carotenoids called xanthophylls. They have one hydroxyl group and are more polar than carotenes [16]. These carotenoids are thought to be effective at absorbing damaging blue light entering the eyes [25]. This is in addition to their potent antioxidant properties [4,26]. Carotenoids are, in general, not water-soluble, but are very fat-soluble compounds, and in the blood stream are transported by the low-density lipoproteins (LDL) and high-density lipoproteins (HDL) [23,24], which in turn are bound by albumin [24,25,26,27]. Serum albumin is the predominant protein found in the blood stream and is responsible for transporting important substances, such as hormones and carotenoids, through the body [27,28].

This study seeks to develop the methodology, based on Raman spectroscopy, to critically analyse and compare three important carotenoids: Lutein, Zeaxanthin, and Beta Carotene in serum albumin complexes, with the aim of differentiating their spectral signatures, and quantifying their relative contributions in admixtures. As a substitute for human serum, bovine serum albumin (BSA) was employed, which has been shown to solubilise and protect carotenoids from photodegradation [29,30]. The concentration of BSA was chosen to be physiologically relevant (40 mg/mL serum albumin) [29,30], and that of the carotenoids, over a wide range of 0.05–1.0 mg/mL, was chosen to include ranges of both physiological relevance [31,32] and clinical centrifugation and filtration standards [33]. The study employs four different laser wavelengths (785 nm, 660 nm, 532 nm, and 473 nm) to establish the optimum source wavelength and spectral analysis range for the compounds and explores a range of multivariate statistical analysis techniques for differentiation and quantification.

## 2. Results

### 2.1. UV/Vis Spectroscopy of Carotenoid: BSA Complexes

Carotenoids normally absorb light in the visible region of 400–500 nm and have a long wavelength maximum absorbance in ethanol solution in the region of 450 nm [22,34,35]. Absorbance measurements carried out for BSA complexes of the three carotenoids all show shifts in absorbance towards the red region of the absorption spectrum. Beta Carotene had the biggest shift, exhibiting a longest wavelength absorption feature at 540 nm (Figure 1A). This was followed by Zeaxanthin, which had a similar absorbance feature at 528 nm. Lutein was the least affected, with only a minimal shift and a longest wavelength absorbance feature at 486 nm.

As a result of the spectral shifts, the carotenoid BSA samples had strong orange colour, rather than the typical yellow colour of human serum, as shown in the Appendix A. Similar shifts in the absorbance features of carotenoid compounds have been extensively explored and reported [20,36,37,38,39,40], and can be caused by factors such as aggregation and solvatochromism [41], as a result of changes to the local electronic environment of each molecule due to intermolecular interactions. The degree of aggregation can also influence the molar absorbance values, as evidenced, for example, by studies of Beta Carotene in ethanol:water solutions, in which the peak absorbance was seen to decrease significantly as the ethanol content was reduced to <67% [38], and therefore, although the measured solutions of the three carotenes in Figure 1A were prepared at the same concentration of 1 mg/mL, the resultant absorbance maxima are quite different.

To validate the method employed and the solvent used in preparing the carotenoids, the concentration dependence of the absorbance of Beta Carotene in BSA was examined over a range of concentrations (0.05–1.0 mg/mL) [31,42]. Although the spectral shape evolves somewhat in the range ~400–450 nm, the longest wavelength absorption feature remains at ~540 nm over the concentration range (Appendix A). Figure 1B shows the concentration dependence of the 540 nm absorption maximum of Beta Carotene in BSA over the concentration range. The linear dependence is an indication of good solubilisation of the compound in aqueous solutions of BSA over a well-varied concentration range. Note that, although spectra of all compounds in Figure 1 were measured using a BSA solution as reference, a significant scattering background to the spectra was observed. Appendix A illustrates that the background (typical of Mie-like scattering [43]) is slowly increasing from high to low wavelengths over the spectral range. For the purpose of better visualising the absorption features, the value of the background at 700 nm has been subtracted from the whole spectral range in Figure 1A. This scattering background is also seen to be concentration dependent, although more so at lower concentrations, as shown for the range of Beta Carotene solutions in Figure 1C. If it is assumed that the background has origin primarily in scattering from the BSA protein [44], it may be concluded that complexation with, and aggregation of, the carotenoids increase this scattering.

### 2.2. Raman Spectroscopic Analysis of Beta Carotene/BSA Complexes

Raman spectroscopic analysis of the carotenoids was first carried out on Beta Carotene, as an important and widely used example of carotenoids in the body. Extended multiplicative signal correction (EMSC) was used to perform background correction [45]. Before deciding on the most efficient laser wavelength to be employed, all the components of the BSA/carotenoid complexes were compared. Figure 1A illustrates that the carotenoid complexes have different absorbances at the four-source wavelengths employed in the Raman investigations for this study. The blue (473 nm) and green (532 nm) regions show high absorbance, the green being close to the longest wavelength absorbance maximum, and thus the Raman scattering process is expected to be resonantly enhanced, but also subject to self-absorption. The red (660 nm) and near-infrared (785 nm) show much lower absorbance on the carotenoid/BSA spectrum and are thus near- and/or off- resonant [14].

Figure 2A shows the spectra of the pure Beta Carotene compound (paste) at the four different source wavelengths. This was used as reference for the EMSC correction. For all wavelengths, the spectra clearly exhibit the characteristic carotenoid peaks at ~1519 cm^−1^, ~1158 cm^−1^, and ~1004 cm^−1^. The relative strengths of these features will depend, however, on the resonance conditions of the source wavelength, relative to the material absorption (Figure 1A) [22]. Note that the spectra have not been corrected for the differing instrument responses, including source power, and responses over the different spectral ranges [46], and therefore, the as-measured relative responses cannot be readily interpreted in terms of the absorption spectra of Figure 1A. However, the uncorrected spectra better represent the quality of the spectral signal collected, to guide the choice of the optimum measurement wavelength. Figure 2B is the water spectrum, used in the EMSC algorithm to subtract water which was part of the background. Water has characteristic peaks at ~1640 cm^−1^ (H-O-H scissors), and a broad band from ~3000–3600 cm^−1^, (OH symmetric and antisymmetric stretch), as can be seen in Figure 2B. Figure 2C is the spectrum of BSA at all four wavelengths, also to be subtracted from the Beta Carotene complex using EMSC. BSA exhibits typical protein features in the fingerprint region of 400–1800 cm^−1^, such as the sharp phenyl alanine peak at ~1004 cm^−1^ and the broad Amide I band, which has its maximum at ~1650 cm^−1^ [47], although the feature also contains several sub-bands, indicative of the protein secondary structure content [48], and in the high wavenumber region, in which the CH_2_ and CH_3_ stretching vibrations feature strongly at ~2900 cm^−1^.

In Figure 3, the Raman spectra of Beta Carotene/BSA complex solutions (1 mg/mL), recorded at 473 nm, 532 nm, 660 nm, and 785 nm, are compared. At this stage (Figure 3A), the spectra contained have been smoothed using the Savitzky-Golay method (polynomial order of 5, window of 9). The smoothed spectra at all wavelengths are dominated by features from Beta Carotene, represented by the three peaks at ~1004 cm^−1^, ~1158 cm^−1^ and ~1519 cm^−1^. The features of water (broad peak at 3000–3600 cm^−1^) and BSA (peak at 2900 cm^−1^) are very prominent in the 532 nm and 473 nm spectra and almost extinct in the 785 nm and 660 nm spectra. Note, however, that the spectra have not been normalised for the instrumental response, which reduces at longer wavelengths, for example because of the diminished response of the silicon CCD. The three major peaks of the carotenoids, which come from the carbon-carbon double and single bond stretches of the polyene backbone and methyl bends of the carotenoid structure, are best highlighted in the 532 nm spectrum, followed by the 785 nm spectrum. The 1519 cm^−1^ peak corresponds with the *V*_1_ band which originates from vibrations of conjugated C=C stretching modes of the double bonds [49,50]. The 1158 cm^−1^ peak corresponds with the *V*_2_ band which originates from a combination of C=C and C-C single bonds stretching with C-H bending modes [25,50]. The 1004 cm^−1^ peak corresponds to the *V*_3_ band that derives from C-CH_3_ bonds found between the main carbon chain and the side methyl groups in the carotenoid molecule [25,50]. There is a small band at 950 cm^−1^, known as the *V*_4_ band, associated with out of plane C-H modes. The other smaller bands which contribute to the Raman spectra are resonantly enhanced in the 532 spectra. These weaker features, such as those at ~2126 cm^−1^, ~2679 cm^−1^, and ~3036 cm^−1^, are overtone and combination bands which arise from the total symmetric character of the carotenoid vibrations (*V*_1_, *V*_2_ and *V*_3_) [51,52]. They also appear to be extinct or very poorly visible on the 785 nm, 473 nm, and 660 nm spectra but are all quite prominent in the 532 nm spectrum.

After background subtraction and normalisation for the water contribution (Figure 3B), the 532 nm spectrum of Beta Carotene (green) clearly stands out, having the strongest and best-defined carotenoid features. It highlights the three major peaks most prominently but also all the minor features of the spectrum that can easily be missed using the other lasers. This is most likely because of the resonance effect on the scattering of Beta Carotene [50,53] using the 532 nm laser as source. The results indicate that 532 nm is the best choice for accurate and optimum qualitative and quantitative analytical deductions of Beta Carotene and other carotenoids solubilised in albumin and may, therefore, serve as a basis to quantify other carotenoids/BSA complexes.

In Figure 4A, the as-measured Raman spectra of Beta Carotene (532 nm) in BSA solutions of concentration ranging from 0.05 mg/mL to 2 mg/mL over an acquisition time of 4 s and an accumulation of 5 are shown, over the range 400–3100 cm^−1^. Comparison of the multiple spectra from the range of samples and measurements illustrates that the spectrum is quite variable, in terms of background, and relative contributions of the typical carotenoid features and the water features at ~3200 cm^−1^. After applying the Savitzky-Golay algorithm, the figure shows less noisy and smoothened spectra. At this stage, the spectral profile is a mix of signals from water, BSA, Beta Carotene, and stray light scattering. The EMSC algorithm was used to improve the spectra by performing background correction. Water constituted the inherent background and were subtracted simultaneously by imputing their known (reference) spectra (Figure 2B) and BSA (Figure 2C) into the EMSC algorithm. A polynomial of 7 was chosen as the best order for correcting the baseline. Figure 4B shows the spectra after subtracting the inherent background from the raw spectra, but without water normalisation. It was seen that the water subtraction protocol, on occasion, resulted in negative features in the region 3200–3800 cm^−1^, and therefore, the corrected spectra are displayed in the region 400–3100 cm^−1^. Further investigation would be required to investigate whether these may have origin in slight changes to the water spectrum as a result of interaction with the solute. The spectra are now free from interferents from water and BSA, and exhibit the clearly distinct Beta Carotene peaks, three of which dominate the spectra and a few other smaller peaks at 2169 cm^−1^, 2526 cm^−1^, 2679 cm^−1^, and 2920 cm^−1^. Figure 4C shows the corrected spectra after water normalisation. Water is the predominant constituent of serum, and as the same volume of sample is measured on each occasion, the water content can be used as an internal standard to normalise the measured responses, minimising measurement variability. The least-squares fitting of the EMSC algorithm identifies the water contribution to each spectrum, and the weighting co-efficient can then be used to normalise each spectrum. This extra step brought about a visible improvement in the spectral variability, as evidenced by the concentration dependence of the 1519 cm^−1^ peak of Beta Carotene before (Figure 4D) and after (Figure 4E) water normalisation. Note the different scales of Figure 4D,E, due to the normalisation of the data by the EMSC least-squares fitting parameter for water.

Figure 5 shows the concentration dependence of the most prominent peak (1519 cm^−1^) of the Raman spectrum of Beta Carotene BSA solution using 532 nm as source, before and after self-absorption correction over the concentration range of 0.05 mg/mL to 2 mg/mL. While this should increase linearly as a function of concentration, when the solution reaches a concentration of ~0.6 mg/mL, what looks like saturation appears to occur. A similar behaviour is observed for the features at 1158 cm^−1^ and 1004 cm^−1^ (data not shown). It should be noted that the source wavelength of 532 nm is strongly absorbed by the solution itself, as is the Stokes-shifted, Raman back-scattered light, on its return to the objective. This self-absorption can be corrected for by the algorithm of Lu et al. [22], described in Section 4.5. The *I_Rm_* data of Figure 5A was fitted using a problem based nonlinear-least-squares algorithm, based on Equation (3), using the values of α*_L_* (532 nm) and α*_R_* (579 nm, corresponding to a Raman shift of 1519 cm^−1^), yielding values of *d*_1_ = 1.3 and *d*_2_ = 1.75. These values were then used to calculate the original Raman scattering, *I*_*R*0_, using Equation (4), resulting in an improved linearity (Figure 5B). The results confirm the linearity observed for the UV/Vis spectra in Figure 1B and give further confidence that quantitative analysis of the Carotenoid/BSA complex solutions can be performed.

### 2.3. Raman Spectroscopic Analysis of Carotenoid/BSA Complexes: Multivariate Regression Analysis

PLSR against concentration was carried out on the concentration-dependent spectral data of Beta Carotene/BSA complex solutions, to construct a regression model using their Raman signals. A model was first constructed using 80 components (or Latent Variable, *N* = 80 corresponding to the number of spectra minus 2). The cumulative percentage variance as a function of number of latent variables was calculated (Figure 6A), indicating that 8 LVs were sufficient to account for 100% of the variance in the dataset, and the model was reconstructed accordingly.

Figure 6B shows the predictive model constructed using 8 components, and a 10-fold cross validation, showing an appreciable linearity (R^2^ = 0.9995) between the Predicted and Measured responses, which correspond to the input target values of Beta Carotene concentration. The spectrum of the first LV reproduces that of the Carotenoid (not shown), and, as expected for a well-correlated model, the PLSR Co-efficient is also dominated by the features of Beta Carotene (Figure 6C). Figure 6D shows the dependence of the Mean Squared Cross Validation error on the number of LVs used. In fact, the analysis shows that only two components are enough to construct the model, as the RMSE of prediction is minimised after 2 LVs, consistent with the fact that the spectrum of the first LV dominates the PLSR Co-efficient. The PLSR model indicates that an unknown concentration of Beta carotene in BSA solution, within the range 0.05–2.0 mg/mL, can be predicted with an accuracy of RMSECV = 0.0032 mg/mL. The limit of detection can be estimated as LOD = 3.3 × RMSECV = 0.0106 mg/mL [54].

### 2.4. Quantitative Analysis of Beta Carotene, Lutein and Zeaxanthin

Having demonstrated quantitative analysis for Beta Carotene as a key example of the carotenoids, the Raman spectroscopic analysis was extended to all three carotenoids in BSA using 532 nm as source, as has been shown to give the best results in the initial analysis of Beta Carotene. 

Figure 7A shows the as-measured 532 nm spectra of Lutein, Beta Carotene, and Zeaxanthin in BSA solution, each at a concentration of 1 mg/mL. Due to the resonant conditions, the carotenoid peaks dominate over the features of the BSA and water. Figure 7B is the spectra of all three carotenoids after subtracting the background water and BSA, and normalising for the water content, while Figure 7C shows the spectra after correction for self-absorption, in each case using the respective absorbance spectrum of Figure 1, although also including the background scattering. Self-absorption reduces the laser intensity as it penetrates the sample, and also the amount of Raman scattered light as it emerges from the sample. Correction for this therefore increases the representation of the Raman Intensity, compared to the as-measured spectrum. After correction, the spectrum of Lutein is seen to be strongest, while that of Beta Carotene is weakest. Although the Raman cross section (scattering) is expected to be similar for each compound, the resonant enhancement at 532 nm is different for each (Figure 1A), Zeaxanthin being the highest, and Lutein the lowest. The degree of vibrational damping due to aggregation will also influence the spectral responses, and so the differing spectral intensities of Figure 7C are not unsurprising.

### 2.5. Differential Analysis of Beta Carotene, Lutein and Zeaxanthin

To successfully use Raman spectroscopy to quantify the carotenoids in admixtures of various concentrations in serum, it is important to consider whether the spectra of Beta Carotene, Lutein, and Zeaxanthin can be discriminated. A closer examination of the spectra of Figure 7C indicates that the strongest feature at ~1519 cm^−1^ is shifted to lower wavenumbers in Beta Carotene, and this shift is greatest relative to Lutein, for which the feature is positioned at a higher wavenumber than for Zeaxanthin (Figure 7D). The other major peaks at ~1159 cm^−1^ and ~1004 cm^−1^ are observed to be relatively stronger in Beta Carotene, and similar for Lutein and Zeaxanthin, while the features > 2000 cm^−1^ appear to be strongest, and also shifted, in the Raman spectrum of the Zeaxanthin complexes (Figure 7E).

These subtle spectral differences suggest the potential to quantitatively evaluate relative contributions of specific carotenoids in mixtures. In order to examine the variability of the spectral differences, PCA was used to make detailed comparison of the Raman spectra taken from all three carotenoids by looking at variations within the dataset of all three complexes. The dataset for all three complexes was vector normalised before analysis, and the PCA algorithm includes a mean centering process.

Figure 8A shows the %Variance Explained as a function of the Number of Principal Components, indicating that PC1 accounts for 83.61%, and PC2 14.27%. The scatter plot of PC2 vs. PC1 (Figure 8B) scores shows precise clustering of the spectra of each carotenoid compounds, and a clear differentiation between them. The spectra of Beta Carotene cluster on the negative side of PC1, while those of Lutein cluster on the positive side. The Zeaxanthin cluster appears roughly around the zero of the PC1 axis. Therefore, PC1 primarily differentiates Beta Carotene and Lutein. Figure 8C shows the spectral loading of PC1, in which the biggest variance from the carotenoids is seen at the 1519 cm^−1^ peak. The variance comes from both the negative (Beta Carotene) and positive (Lutein) sides of PC1. There is also variance in the 1004 cm^−1^ peak as well as the 1154 cm^−1^ peak, which mostly comes from the negative side (Beta Carotene) of the loading, indicating that this feature is stronger in Beta Carotene than in Lutein. Minor variances can be seen in the smaller peaks at ~2169 cm^−1^, ~2526 cm^−1^, ~2679 cm^−1^, and ~2920 cm^−1^ on both the positive and negative sides of the loading. PC2 differentiates Zeaxanthin from the other two carotenoids, and its loading (Figure 8D) is seen to exhibit variances across the spectrum. Again, the biggest variance comes from the 1519 cm^−1^ peak, the derivative line-shape indicating a shift in the peak in Zeaxanthin relative to the other two carotenoids. The two other major peaks have variances mostly on the negative side (Beta Carotene) of the loading and the minor peaks are also very varied on the opposing sides. 

In the pairwise analysis of the carotenoids, Lutein and Zeaxanthin spectra are clearly differentiated, with a variance of 95.66% from PC1 (Figure 9A). There is a tight clustering of the Lutein spectra, which is seen on the positive side of PC1. The Zeaxanthin spectra, on the other hand, appear to be more dispersed through the negative and positive sides of PC2 and concentrated on the negative side of PC1. The spectral loading of PC1 for Lutein and Zeaxanthin (Figure 9B) shows the variance from both carotenoids is mostly from the 1519 cm^−1^ peak with very little variance observed from the other two major peaks. The derivative-like line-shape of the PC1 loading indicates a shift of this peak from a slightly higher wavenumber in Lutein to lower in Zeaxanthin. The negative peaks in the region > 2000 cm^−1^ indicate that these features are relatively stronger in Zeaxanthin.

When Beta Carotene and Zeaxanthin are paired in the scatter plot (Figure 9C), PC1, which accounts for 94.89% of the variance, clearly differentiates the two carotenoids, with Beta Carotene on the negative side and Zeaxanthin on the positive. The PC2 accounts for only 1.64% of the variance and the spectra from both carotenoids are spread through the positive and negative sides of the loading, with Zeaxanthin spectra most varied. In the PC1 spectral loading plot for Beta Carotene and Zeaxanthin (Figure 9D), the 1519 cm^−1^ peak is the most varied for both carotenoids. The variation on the other two major peaks is seen mostly from the negative side (Beta Carotene) of the loading. In this case, the derivative-like line-shape of the PC1 loading indicates a shift of this peak from a slightly higher wavenumber in Zeaxanthin to lower in Beta carotene. The negative peaks in the region < 1500 cm^−1^ indicate that these features are relatively stronger in Beta Carotene, while the positive peaks > 2000 cm^−1^ indicate that these features are relatively stronger in Zeaxanthin.

In a pairwise PCA of Beta Carotene and Lutein, the carotenoids were clearly differentiated in the scatter plot. PC1 accounted for 98.36% of the variance while PC2, 0.53% of the variance (Figure 9E). There is a tight clustering of the Lutein spectra when compared to the Beta Carotene spectra which is more dispersed through the negative and positive sides of PC2. In the PC1 spectral loading plot (Figure 9F), a similar result to that of the Beta Carotene and Zeaxanthin loading plot is seen. The derivative-like line-shape of the PC1 loading indicates a shift of this peak from a slightly higher wavenumber in Lutein to lower in Beta carotene. The negative peaks in the region < 1500 cm^−1^ indicate that these features are relatively stronger in Beta Carotene, while the positive peaks > 2000 cm^−1^ indicate that these features are relatively stronger in Lutein, and, in this case the derivative-like line-shape of these features indicates a slight shift to lower wavenumbers in Beta Carotene.

### 2.6. Quantitative Analysis of Beta Carotene, Lutein and Zeaxanthin

Having demonstrated that the carotenoids can be effectively discriminated, quantitative analysis of Beta Carotene, Lutein, and Zeaxanthin was further carried out in four different admixture ratios and the Raman spectra analysed using problem-based, nonlinear least squares fitting to quantify the individual components contained in the mixtures. The fitting function included correction for the self-absorption due to each component of the admixture, and the fitting parameters were the relative concentrations of each of the three carotenoid/BSA complexes.

Figure 10A shows the example of the measured and fitted spectra of an admixture of Beta Carotene:Lutein:Zeaxanthin in the ratio 100:50:10. In general, the best fit was obtained using the full spectrum of 400–3100 cm^−1^, and reducing the fitting range to <2000 cm^−1^, >2000 cm^−1^, or 1350–1750 cm^−1^ did not improve the fitting. Fitting of second derivative spectra similarly did not improve the quality of the fitting. Fitting the spectra of each of the individual carotenoid/BSA complexes yielded a 100% contribution of the respective complex. In the example of Figure 10A, a fitted ratio of 96:51:13 was obtained, although for multiple measurements of admixtures of ratios, 100:50:10, 100:40:20, 100:30:20, 100:20:40, the fitted ratios were somewhat variable, as shown in the plot of fitted % versus nominal % in Figure 10B. Nevertheless, a reasonably good correlation between fitted % versus nominal % is observed. A linear regression of the full dataset yielded an R^2^ value of 0.78, and a Standard Error of 13.7%. For Beta Carotene, an R^2^ value of 0.33, and a Standard Error of 15.3% were found, although only two values of nominal % were employed (100% and 62.5%), and so a linear regression is not so appropriate. Similar analysis for Lutein (100% to 12.5%) yielded an R^2^ value of 0.65 and Standard Error of 16.3%, and for Zeaxanthin (100% to 6.25%), an R^2^ value of 0.98 and Standard Error of 3.4%.

## 3. Discussion

The potential for Raman spectroscopy to provide accurate predictions of molecular constituents in complex biological media is increasingly being explored [47,55]. Previous studies have thoroughly described the use of Raman to detect/analyse carotenoids in foods and bodily tissues [26,56,57]. This study explored the sensitivity and specificity of Raman spectroscopy for the quantitative analysis of key carotenoids solubilised in serum albumin, as a model for blood serum, rather than organic solvents, commonly used in carotenoid studies [22,49,58]. Serum albumin was chosen because carotenoids are naturally bound to albumin in the blood stream, so it was an ideal and inexpensive model in which to examine the potential of Raman spectroscopy to differentiate and quantify carotenoids of potential diagnostic interest. In all measurements, the serum albumin concentration was ~40 mg/mL, consistent with physiological levels [59,60].

Beta Carotene is the most widely studied carotenoid in the human body [23] and it was therefore chosen to establish and demonstrate the methodologies of the generation of serum albumin complexes and their characterisation using UV/visible absorption and Raman spectroscopies. As the spectral properties of other dietary carotenoids are very similar, the deductions from the analysis of Beta Carotene can be applied to them. The absorbance spectra of Beta Carotene measured in the visible range was seen to have a longest wavelength absorption maximum at ~540 nm, considerably red-shifted from that reported in ethanol solutions [38,39,40], consistent with aggregation in the BSA complex environment. The longest wavelength absorption maximum of Beta Carotene has been shown to be shifted from ~476 nm in ethanol to ~515 nm in 1:1 ethanol water solution due to J aggregation [35,38,39,40]. Similar spectral shifting in a range of solvents has been documented, depending on polarisability [37,61,62], and the UV-Vis spectrum of Beta Carotene in solution is also pressure-dependent, the longest wavelength absorption maximum shifting as low as ~580 nm in carbon disulphide solution at 0.96 GPa [63]. In H aggregates of Zeaxanthin, prepared in Ethanol:Water mixtures, the UV-vis spectrum is dominated by a strong feature at 370 nm, while in 1:9 THF:Water solutions, the longest wavelength absorption maximum is shifted from 485 nm to ~510 nm [64,65]. A similar red shift is seen in J aggregates of Lutein [66]. The spectral profile is consistent over the concentration range, further verified by the linear concentration dependence of the absorption maximum (Figure 1C).

Comparing the four different laser wavelengths employed in this study, 532 nm appeared to be the best wavelength for the Raman analysis, as it benefitted from a strong instrument response across the spectral range and was on or near resonance enhancement, consistent with previous studies [7,53] which illustrate the use of resonance Raman as a key advantage in studying carotenoids. In the Raman spectroscopic analysis of Beta Carotene, the three dominant characteristic Raman features of the carotenoid can be seen clearly even in the unprocessed spectra, which represent important details in the carotenoid structure [49,50]. The Raman spectrum sits on a large broadband background which, because carotenoids exhibit negligible fluorescence emission [67], is attributable to stray light due to scattering of the cloudy BSA solution [68]. The EMSC algorithm was able to remove the contributions of this background, as well as the interferents (water/BSA) and the corrected spectra clearly highlight the Raman spectral characteristics of the carotenoids only. Notably, normalisation for the water subtracted before further analysis helped to reduce measurement variability, and, once normalised, the spectra should be quantitatively comparable. After pre-processing, as well as the dominant features at ~1519 cm^−1^, ~1158 cm^−1^, and ~1004 cm^−1^, prominent features were observable in the region 2000–2700 cm^−1^, which contributed to the differentiation of the carotenoids. Notably, CH or OH features in the region > 2900 cm^−1^ are not resonantly enhanced, and thus any characteristic carotenoid features are obscured by those of the protein/water environment.

The concentration dependence of the Beta Carotene/BSA response clearly shows, however, the influence of self-absorbance, which can be different for different carotenoids. The corrected concentration dependence (Figure 4B) shows a consistent increase in intensity as the concentration increased which suggest that Raman spectroscopy of carotenoid/BSA complex can be quantitatively analysed.

The PLSR model of the concentration dependent Beta Carotene spectral data showed a strong correlation between the concentration of Beta Carotene solution and the Raman spectral profile. The analysis was carried out over a wide range of low to very high concentrations, to intensify the resonance Raman signals obtained from the 532 nm laser which helped in the differentiation of the components. The mean physiological levels of Beta Carotene in serum are 2.5 × 10^−4^ mg/mL [32], although they can increase by factors of up to x 4–6 due to dietary supplementation [69,70,71,72]. Extracting the high molecular weight fraction of serum for clinical analysis by centrifugal filtration increases the albumin concentration, and therefore carotenoid concentration bound to it, by a factor of ~10 [33]. The concentration range used for the Raman spectroscopic study is therefore of physiological relevance, and the RMSECV of 0.0032 mg/mL indicates the technique may be developed for quantification of carotenoid content of human serum.

Although the spectra look very similar, PCA was chosen for the differential analysis of the spectra of Beta Carotene, Lutein, and Zeaxanthin, to explore subtle differences. The analysis clearly indicated shifts in the spectral positioning of the strongest peak at ~1519 cm^−1^, to higher frequencies, in the order Beta Carotene, Zeaxanthin, Lutein, as well as similar spectral shifts and intensity differences in the combination and overtone modes in the region 2000–3100 cm^−1^. This rather impressive outcome of the differential analysis provided the basis on which the quantification analysis was carried out in admixtures. A 5:3 ratio of Beta Carotene to Lutein + Zeaxanthin was chosen for physiological relevance [32,73], and the ratio of Lutein:Zeaxanthin was varied from 5:1 to 1:2. The fitting was undertaken over the full spectral range, which was found to be the best, especially because the results of the differentiation showed variances in the high as well as the low wavenumber regions. The program was first tested by fitting it with the individual carotenoid/BSA components, which yielded 100% for each of the components. An example of the approximate carotenoids ratio in the blood (100:50:10) [74] was used to illustrate the result of the fitting (Figure 10B). The actual admixtures gave somewhat variable contributions for the carotenoids when fitted, although the results are reasonably correlated with the ratios of the as-prepared admixtures.

Carotenoids’ content in serum is known to be very small amongst other biological constituents/components such as urea, glucose, globulins, etc. [42,47] Studies have previously shown that Raman spectroscopy can distinguish the high and low molecular components in the serum, and quantify the concentrations of the constituent components with a high degree of precision [47,75]. The results from this study indicate that the technique can potentially be used, not only to quantify the carotenoid content of human serum, but also to potentially identify the differential contributions of different carotenoids of potential importance for clinical analysis of carotenoids implicated in diseases of the eye (where Lutein and Zeaxanthin are especially abundant) and other carotenoid-related disorders.

## 4. Materials and Methods

### 4.1. Sample Preparation

Beta Carotene, Lutein, and Zeaxanthin powders were all purchased from Sigma Aldrich (Arklow, Ireland). Powders were dissolved in 40 mg/mL BSA (Sigma Aldrich, Arklow, Ireland) to give a final concentration of 1 mg/mL. The stock solutions were prepared using ultra-pure water (Millipore) and used immediately to prevent oxidation under light and air.

The carotenoids (Beta Carotene in particular) were seen to be poorly soluble in BSA solution, and mild sonication using a sonic VCX—750 Vibra cell ultra-sonic processor (Sonics and materials Inc., Newtown, CT, USA), equipped with a model CV33 Sonic Tip sonication probe (20% amplitude for 10 s) was used to disperse the carotenoid solids more evenly in the BSA solute. To confirm linearity of the solubilisation of the carotenoids in BSA, a range of different concentrations of Beta Carotene were generated. Several dilutions were prepared in BSA with concentrations of Beta Carotene ranging from 0.05 mg/mL to 2.0 mg/mL while keeping the concentration of BSA constant.

Admixtures of the three different carotenoids were prepared by mixing BSA solutions of Beta Carotene, Lutein, and Zeaxanthin in (B:L:Z) ratios reflective of physiological relevance [32,74] as follows: 100:40:20, 100:30:30, 100:50:10, and 100:20:40. The concentration of BSA was kept constant at 40 mg/mL.

To make the carotenoid paste (carotenoid reference for Section 4.4), 20 uL of ultrapure water was added to 1 g of Beta Carotene powder and mixed until a thick paste was formed.

### 4.2. Absorbance Measurement

UV-VIS absorption spectra were recorded in the visible range of 400–700 nm using a plate reader, the SpectraMax M3 (Molecular devices). The carotenoids/BSA complexes and admixtures were all measured using a 96-well plate at a fixed concentration to ascertain their absorbance. The control (BSA) was also measured. To examine the concentration dependence, various concentrations (2 mg/mL to 0.05 mg/mL) of Beta Carotene in BSA solution were measured, while maintaining the concentration of BSA constant throughout. Absorbance of the solution at 540 nm was plotted against concentration.

### 4.3. Raman Analysis

Raman spectral measurements were carried out using a Horiba Jobin-Yvon LabRam HR800 spectrometer with a 16-bit Peltier cooled CCD detector, coupled to an Olympus BX41 upright microscope. The laser lines used were 473 nm, 532 nm, 660 nm, and 785 nm, in each case with a 300 lines/mm grating. The spectral range employed was 400–3500 cm^−1^ and the back-scattered Raman signal was typically accumulated for 5 × 4 s. Depending on the measurement, 3–9 spectra were acquired per sample.

Raman measurements of the pure compounds were obtained by measuring a wet paste of the compound at room temperature with a ×60 objective and at the four different laser wavelengths. For the BSA complexes, measurements were performed by focussing into the solutions contained in a polystyrene 96-well plate, using a ×10 objective.

To examine the concentration dependence, various concentrations of the Beta Carotene in BSA solution were further measured at 532 nm while keeping the concentration of BSA constant.

### 4.4. Raman Spectral Pre-Processing

Pre-processing techniques were applied to the raw Raman spectra within the MATLAB platform to correct for excess noise and remove inherent background signals. Smoothing and noise correction were performed using the Savitzky-Golay algorithm [76], using a polynomial order of 5 and a window of 9. Further processing was undertaken using the Extended multiplicative signal correction (EMSC) algorithm [45,47]. Pre-processing was necessary to remove the interferent water and BSA spectra, which made up the solvent in which the carotenoids were dissolved. The reference for the EMSC was obtained for each source wavelength by adding a few drops of distilled water to a known amount of the carotenoid powder and making a thick paste. Using the modification of the process by Parachalil et al., the corrected spectra can be normalised by the co-efficient of the subtracted water, as an internal standard [47,55].

### 4.5. Self Absorption Correction

In the case where the wavelength of the Raman source is resonant with the sample absorption, the intensity is reduced as it propagates through the sample, and the Raman scattered light itself can be similarly attenuated by the sample absorption [22,77]. This self-absorption process can result in a deviation from the linearity of the concentration dependence of the measured Raman signal [78].

The correction method described by Lu and colleagues in 2018 [22] was employed to correct for self-absorption. In accordance with Beer’s law, the depth (*Z*) profile of the Raman scattering intensity *I_R_*(*Z*), including self-absorption, is described by:(1)$$I_{R\;}\left(Z\right)=(I_{R0}e^{-\alpha_{L}Z})e^{\alpha_{R}Z}$$
where *I_R_*_0_ is the intensity of Raman scattering without absorption; *α_L_* and *α_R_* are extinction coefficients at incident laser and Raman scattering wavelengths, respectively [22,79,80]. The measured Raman scattering, *I_Rm_*, can be expressed as:(2)$$I_{Rm.tot}=\int_{0}^{d}\left(I_{R0}\right)e^{\left(\alpha_{L}+\alpha_{R}\right)Z}dZ$$
(3)$$=\frac{I_{R0}}{-\left(\alpha_{L}+\alpha_{R}\right)}\;\left(e^{-}{}^{\left(\alpha_{L}+\alpha_{R}\right)d}-1\right)$$
where *d* is the thickness of solution layer. Correcting for self-absorption, the original intensity of Raman scattering (without absorption) is described by:(4)$$I_{R0}=I_{Rm.tot}\frac{\alpha_{L}+\alpha_{R}}{1-e\left(\alpha_{L}+\alpha_{R}\right)d}$$

The Raman spectrum of a compound can therefore be corrected, knowing the (concentration-dependent) absorption at the source wavelength, and across the Raman spectral range. In the measurements reported here, the measurement pathlengths for UV/visible absorption and Raman spectroscopy are different, and therefore, the exponent in Equation (3) is amended to α*_L_d*_1_ + α*_R_d*_2_.

### 4.6. Partial Least-Squares Regression Analysis and Cross Validation

Following pre-processing and the elimination of inherent background from the spectra, multivariate regression analysis of concentration-dependent Raman responses was carried out using partial least-squares regression (PLSR) to confirm the linear concentration dependent responses, and to demonstrate the predictive capacity of the technique.

The PLSR algorithm looks at the variation in spectral data or predictors, (X matrix), as they relate to the associated factors or responses, (Y matrix), according to the linear equation Y = XB + E, where B is the regression coefficient matrix and E is the residual matrix [75,81]. The Y matrix, or “target” variable is usually a quantifiable or systematically varied external factor, in this case carotenoid concentration. It then attempts to maximise the covariance of X (the Raman spectra) and the target, Y, described according to Latent Variables in a systematic model [75,81]. It can reduce the number of predictors to a much smaller set of uncorrelated components or latent variables which, when summed up, cumulatively and progressively (LV1 > LV2 etc.) account for the co-variance. Least squares regression is therefore carried out on the latent variables, rather than using the original data [81,82]. PLSR can construct a predictive model, which can be used, for example, to predict the value of the target variable, based on the spectrum of an unknown sample, or vice versa.

The loading of the LV reveals the spectral features which contribute to that LV, and therefore to the co-variance. The Regression Co-efficient is the weighted sum of all the contributing LVs, and in spectral analysis, for a good correlation, should yield the spectrum of the constituent components which vary systematically as a function of the target variable.

In the protocol employed, the number of LVs to construct the model was chosen by identifying the point at which the cumulative %Variance Explained reached ~100%. The model was then subjected to a 10-fold Leave-One-Out Cross-Validation process, repeated 100 times, to establish the Root Mean Squared Error of Cross-Validation.

The *K*-fold cross-validation technique was used in this study to validate the model created. This method of cross-validation is a non-exhaustive method, whereby the original dataset is divided randomly into *K* equal subsample sizes. One of the *K* subsamples is used as the validation data for testing the model and the remaining subsample is used as training data. Cross-validation is repeated *K* times and each of the subsamples is used once as the validation data. Starting from the first element, the cycle continues until all the components have been trialed as ‘test’. To improve the number of latent variables used to create the model, the value equivalent to the minimum of the root mean square error of cross validation (RMSECV) and percent variance was estimated. RMSECV is useful to assess the efficiency of the prediction model created while the percentage of variance accounts for the validation of the number of components to be used to obtain the highest variation from the data. When *K* is equal to the number of observations (n), *K*-fold cross validation is the same as the Leave-One-Out Cross-Validation (LOOCV).

In this study, a 10-fold cross-validation (*K* = 10) was carried out. Here, the observation set is divided into 10 equal sizes by random selection. The cross-validation process is then carried out 100 times. During the sequence, each observation is used for testing just once, but all the observations are used for training and testing. An average is obtained from the result, and this is used to produce a single estimation.

### 4.7. Principal Components Analysis

Principal Components Analysis (PCA) is a multivariate analysis technique frequently used in analysing multi-dimensional data sets. It can reduce the number of variables in data sets with multiple dimensions without altering the major variations within the data set [83]. The order of the principal components describes the relevance to the data set. For instance, PC1 should describe the highest variation in the data, followed by PC2, PC3, and so on. The first three PCs will generally provide up to 99% variance in the data set, giving the best visualisation of the differentiation in the cluster sets [83,84].

In this study, PCA was used to explore the differentiation of the carotenoid spectral data sets of Beta Carotene, Lutein, and Zeaxanthin. Data which had been previously corrected for interferents, noise, self-absorption, and normalised for water content were used. The most prominent PC loadings were used to highlight the differences in the data set for the carotenoids.

### 4.8. Nonlinear Least-Squares Curve Fitting

Nonlinear least-squares curve-fitting using a problem-based workflow in the MATLAB environment was employed to undertake the self-absorption correction of the concentration dependence of the Raman spectra of Beta Carotene/BSA complexes, and to fit weighted sums of the constituent spectra to admixture spectra of different ratios, correcting for the self-absorption of each of the constituent components.

In the first case, the function was defined by Equation (2), and the problem was solved by fitting the concentration dependence of the measured Raman signal at ~1519 cm^−1^ by optimising the parameters for *d*_1_ and *d*_2_. Equation (3) was then employed to correct the concentration-dependent spectra of Beta Carotene, as well as the 1 mg/mL spectra of Lutein and Zeaxanthin, for self-absorption.

In the second case, the problem was defined by Equation (3), and the problem was solved by fitting weighted sums of the Raman spectra of the three constituent carotene/BSA complexes, corrected for self-absorption, to the measured spectrum of the admixture, by optimising the weighting parameters, A(1), A(2), and A(3), for each constituent component. 

## Figures and Tables

**Figure 1 molecules-27-04724-f001:**
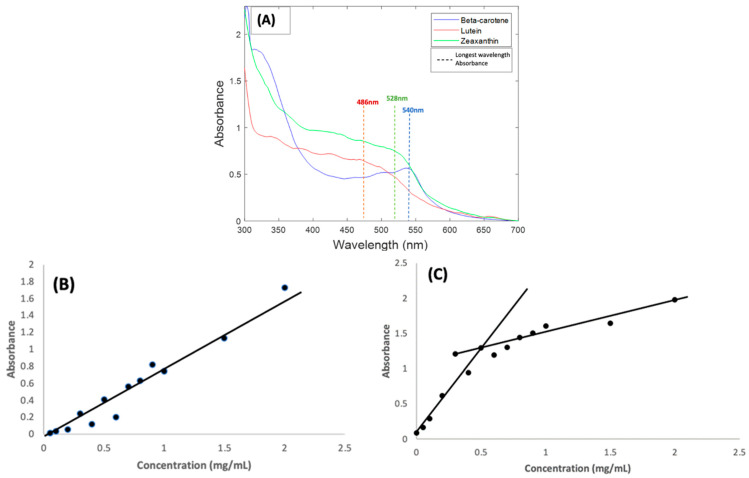
(**A**): Absorption spectra of 1 mg/mL BSA complexes of Beta Carotene, Lutein, and Zeaxanthin solutions. The coloured lines indicate the approximate positioning of the longest wavelength absorption features (**B**): Concentration dependence of Beta Carotene/BSA complex solutions absorbance at 540 nm over the range 0.05 mg/mL–2.0 mg/mL. (**C**): Concentration dependence of Beta Carotene/BSA complex solutions background at 700 nm.

**Figure 2 molecules-27-04724-f002:**
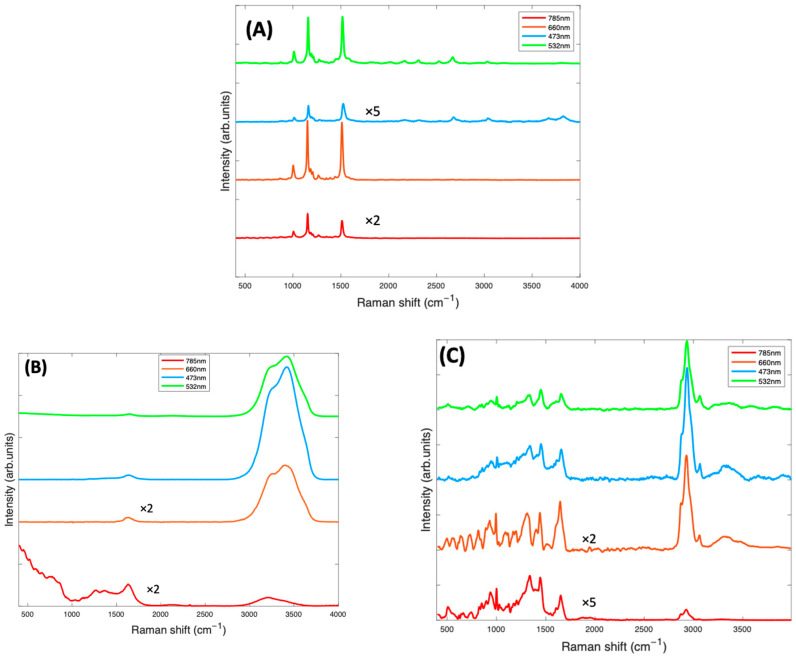
(**A**): Raman spectra at 4 different source laser wavelengths of Beta Carotene reference (Beta Carotene paste) taken with a ×60 objective and spectra of other components of the background contained in the Beta Carotene/BSA complex consisting of water (**B**) and BSA (**C**), both taken with a ×10 objective. The spectra have been offset by a constant factor for clarity.

**Figure 3 molecules-27-04724-f003:**
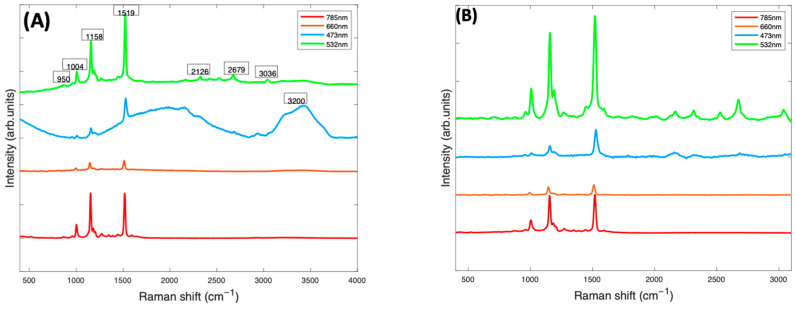
(**A**): Smoothed Raman spectra of Beta Carotene/BSA complex solution (1 mg/mL) at 4 different wavelengths—473 nm (blue), 660 nm (orange red), 532 nm (green), and 785 (red) taken with a ×10 objective. (**B**): Raman spectra of Beta Carotene (1 mg/mL) taken at four different wavelengths after background correction. The spectra have been offset for clarity.

**Figure 4 molecules-27-04724-f004:**
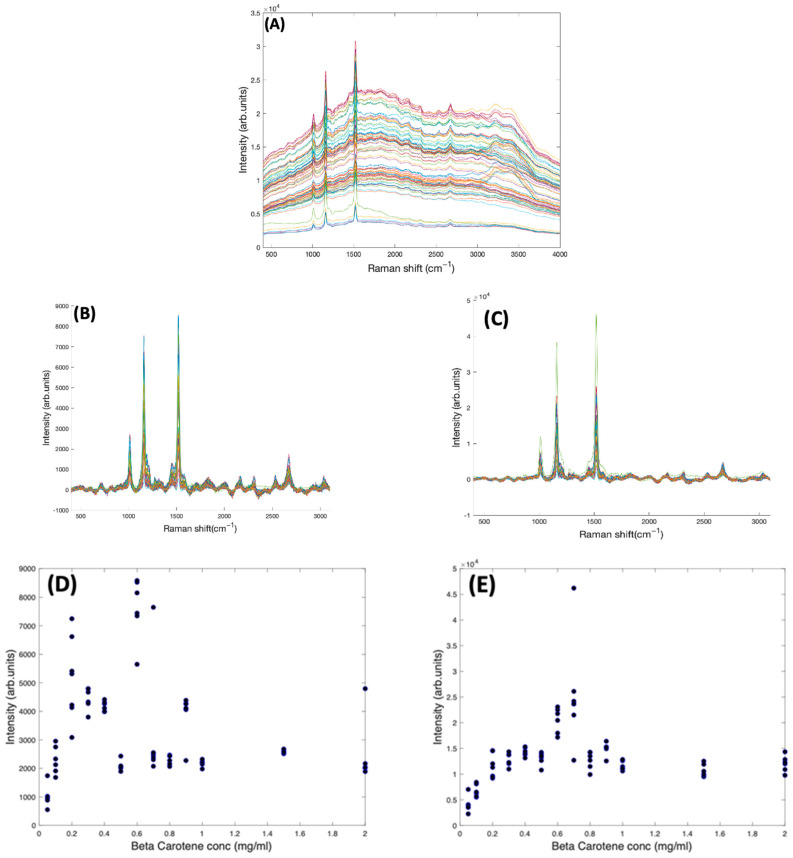
(**A**): Spectra of varying concentrations of Beta Carotene solution (0.05–2.0 mg/mL) showing fairly tightly grouped spectra before background subtraction. (**B**): Spectra of varying concentrations of Beta Carotene after background subtraction and without water normalisation. (**C**): Spectra of varying concentrations of Beta Carotene after background subtraction and with water normalisation. (**D**): concentration dependence of the 1519 cm^−1^ Raman peak of Beta Carotene before water normalisation. (**E**): Concentration dependence of the 1519 cm^−1^ Raman peak of Beta Carotene after water normalisation.

**Figure 5 molecules-27-04724-f005:**
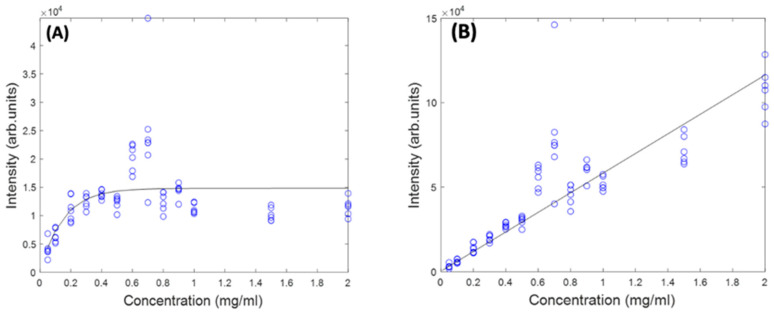
Concentration dependence of 1519 cm^−1^ Raman peak of Beta Carotene at 532 nm before (**A**) and after (**B**) self-absorption correction.

**Figure 6 molecules-27-04724-f006:**
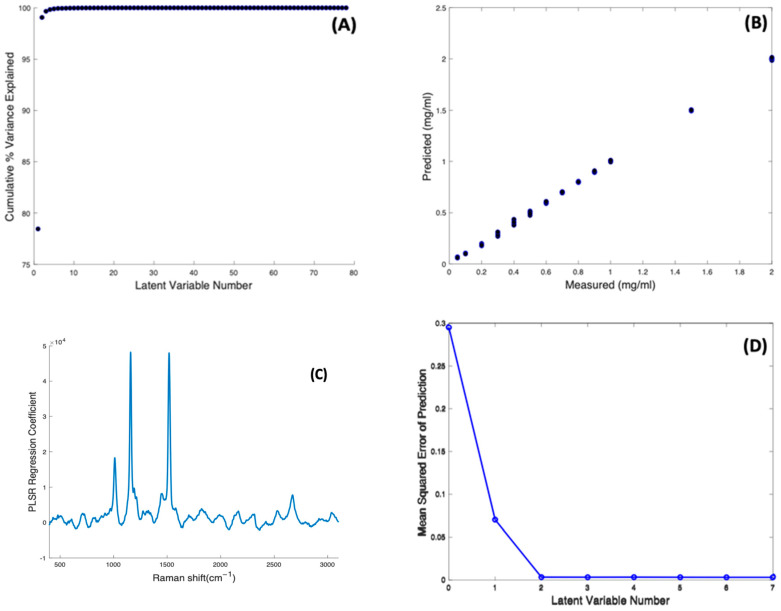
(**A**): Cumulative percentage variance of PLSR using 80 components. (**B**): PLSR predictive model for Raman spectra of Beta Carotene solutions of variable concentrations. (**C**): PLSR regression coefficient of Beta Carotene solution constructed using eight latent variables. (**D**): Estimated mean squared cross-validation error versus number of components (latent variables) for Beta Carotene.

**Figure 7 molecules-27-04724-f007:**
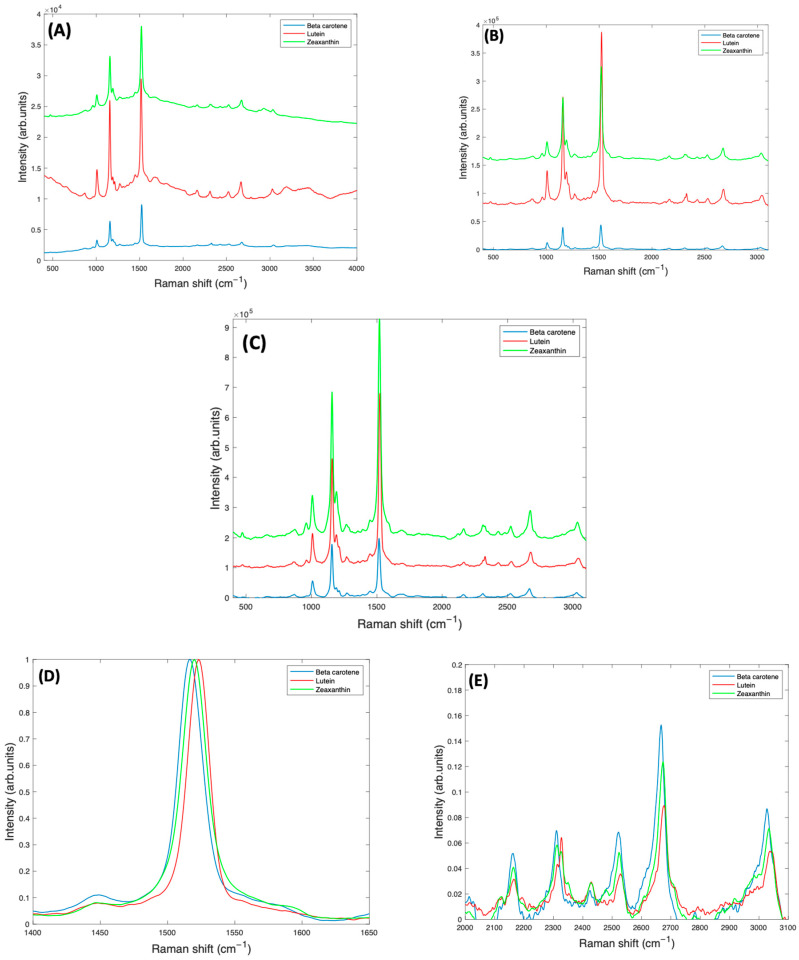
(**A**): Smoothed Raman spectra of 1 mg/mL BSA complex solutions of Beta Carotene, Lutein, and Zeaxanthin, taken at 532 nm, with a ×10 objective lens. (**B**): Corrected Raman spectra of 1 mg/mL BSA complex solutions of Lutein, Beta Carotene, and Zeaxanthin, before self-absorbance correction and offset for clarity. (**C**): Raman spectra of 1 mg/mL BSA complex solutions of Lutein, Beta Carotene, and Zeaxanthin, after self-absorbance correction and offset for clarity. A more detailed comparison of normalised spectra of Beta Carotene, Lutein, and Zeaxanthin is also shown (**D**) in the region of the largest peak at ~1519 cm^−1^ (**E**) in the region > 2000 cm^−1^.

**Figure 8 molecules-27-04724-f008:**
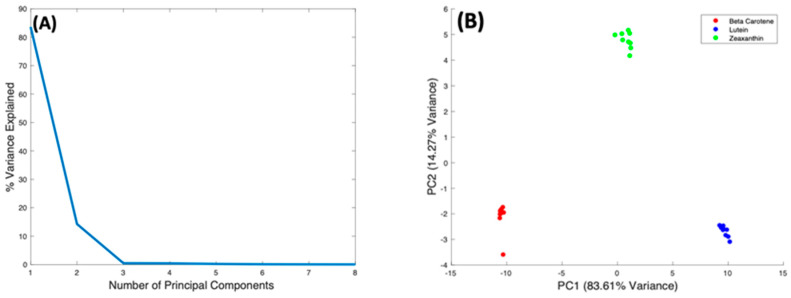
PCA of Beta Carotene, Lutein, and Zeaxanthin. (**A**): Explained percentage variance as a function of number of Principal Components. (**B**): Scatter plot scores for the first 2 principal components. (**C**): Plot of PC1 loading. (**D**): Plot of PC2 loading.

**Figure 9 molecules-27-04724-f009:**
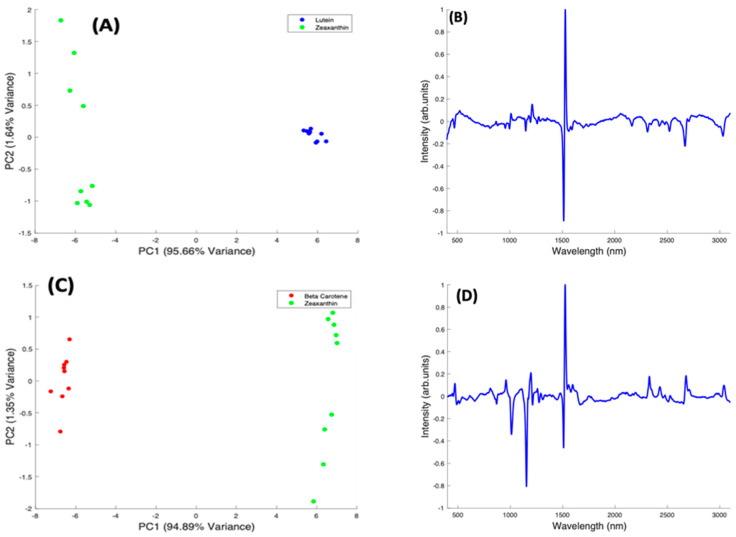
Pairwise PCA analysis for Lutein and Zeaxanthin, and Beta Carotene. (**A**): Scatter plot scores for the first two principal components from PCA of Lutein and Zeaxanthin. (**B**): Plot of PC1 loading for Lutein and Zeaxanthin. (**C**): Scatter plot scores for the first two principal components from PCA of Beta Carotene and Zeaxanthin. (**D**): Plot of PC1 loading for Beta Carotene and Zeaxanthin. (**E**): Scatter plot scores for the first two principal components from PCA of Beta Carotene and Lutein. (**F**): Plot of PC1 loading for Beta Carotene and Lutein.

**Figure 10 molecules-27-04724-f010:**
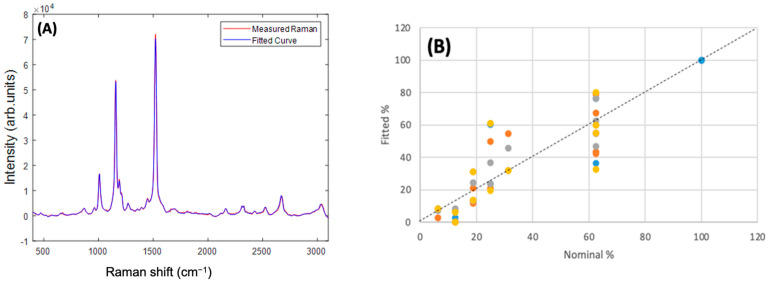
(**A**) Measured and Fitted Raman spectra for the 100:50:10 Beta Carotene:Lutein:Zeaxanthin admixture. (**B**) Fitted% versus Nominal% for each constituent in the range of admixtures. The different colours indicate different replicate measurements.

## Data Availability

Original data is available at https://arrow.tudublin.ie/datas/15 (accessed on 1 June 2022).

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
