# Peer review of "Quantitative Raman Analysis of Carotenoid Protein Complexes in Aqueous Solution"

_molecules, 2022, doi:10.3390/molecules27154724_

Round 1

Reviewer 1 Report

The manuscript submitted by Udensi and collaborators present a work regarding application of Quantitative Raman Analysis of Carotenoid Protein Complexes 2 in Aqueous Solution.

Anyway, I consider if the authors perform significant improvements on the discussion of the results it could be published after some revisions. In addition, there are other aspects that should be properly addressed in a revised version of the manuscript, namely:

·      It was not clear to me the discussion about the displacement of the maximum absorption’s wavelength of the carotenoids. For example, beta carotene has a max absorption’s wavelength at 455 nm (yellow-orange carotenoid), with the results presented by the authors, the authors revealed a new absorption at 540 nm. The yellow-orange beta carotene for example become red???? Please clarify this section, and also include pictures of the samples.

·      The references used are very old, I suggest the authors to update them.

Author Response

Please see the attachment. Responses are colour coded red.

Reviewer 2 Report

The authors reported the use of UV-visible absorption and Raman spectroscopy combined to multivariate statistical analysis to used quantify carotenoid in BSA solution. This study will be the first step before the application of these techniques to identify their differential contributions in clinical analysis. Four different laser wavelengths (785nm, 660nm, 532nm and 473nm) were used to establish the optimum source wavelength that will serve as a basis to quantify other carotenoids/BSA complexes. The authors were then explored the comparison between different multivariate statistical analysis techniques to differentiate and quantify these carotenoids in aqueous solutions.

·         In the abstract (line 17), 532 wavelength excitation is cited twice.

·         Figure 1A showed the absorption spectra of Beta Carotene, Lutein and Zeaxanthin at concentration of 1mg/ml in BSA solution. The authors reported that the maximum absorbance of these three carotenoids shifted towards the red region of the absorption spectrum. The authors should add the absorption spectra of these carotenoids in aqueous solution to confirm this result. In addition, their claimed that Zeaxanthin and Lutein had a maximum absorbance at 528nm and 486nm. By visual inspection of these spectra, no real maximum absorbance were observed for these two carotenoids.

·         In line 106, the authors refer to figure 1A instead of figure 4.1A 

·         The authors reported that these shifts in absorbance towards the red region of the absorption spectrum can be caused by factors like aggregation and solvatochromism. Can the authors explain this point? These spectra were obtained with a concentration of BSA of 40mg/ml serum albumin. The authors should test different concentrations of BSA to confirm this finding. These shifts could be also the consequence of an energy transfer between the carotenoids and the BSA

·         The authors stated that the degree of aggregation can influence the molar absorbance values which can explain the differences in absorbance maxima between these carotenoids. These shifts reported in this section were related to wavelength of maximum absorbance and not to the sample absorbance.

·         Figure 1B shows the concentration dependence of the 540nm absorption maximum of Beta Carotene in BSA over a range of concentrations (0.05 – 1.0mg/ml).  Have the authors checked first whether the increase in the concentration of beta carotene over this  concentration range does not induce a variation in the wavelength of the maximum absorbance of this carotenoid in the BSA due to aggregation and solvatochromism.

·         Scattering background from spectra was removed by subtraction of the absorbance value at 700nm from the whole spectral range. They reported that this scattering background is concentration dependent. The authors should check if this background is wavelength dependent

·         The authors assume that the background has origin primarily in scattering from the BSA protein and aggregation of the carotenoids increases this scattering. The authors should measure the absorption spectra of beta carotene at concentration of 1 mg/mL in different concentrations of BSA to confirm their statements

·         Figure 2 displayed Raman spectra of Beta Carotene reference, water and BSA at four different wavelengths. All spectra presented in this figure should be corrected from the instrument response. In addition except Resonance Raman spectroscopy, Raman spectra (show Raman shifts) should be independent from wavelength excitation.

·         The authors reported that the frequency of amide I band of BSA is located at  1620 cm-1. The authors should check the correct frequency of this Amide I band.

·         Raman spectrum of beta carotene recorded at 660 nm was more intense than that at 532 nm and decreased at 473, and 785 nm. UV-visible absorption spectra (figure 1A) showed high absorbance at 532 and 473 nm. Thus, the Raman scattering process is expected to be resonantly enhanced than those recorded at 660 and 785 nm. Can the authors comment this point?

·         Same remark than figure 1A. The spectra showed in figure 3 should be corrected from instrument response. In figure 3A, the C=C band is located at 1521 cm-1 instead of 1519 cm-1. In addition, they reported that the water and albumin peaks are very prominent in the 532nm and 473nm spectra and almost extinct in the 785nm and 660nm. All spectra did not show any band of BSA. In addition, the absence of water band at high wavenumber region can be explained by the fact that response of CCD detector was reduced at longer wavelengths.

·         In figure 3, the spectra were offset for clarity but no spectrum was multiplied by a factor. The authors should explain clearly the differences in the Raman intensity of beta carotene with respect to the absorbance spectrum of this carotenoid presented in Figure 1

·         In line 219, I did not understand this sentence “Water (Figure 4(B)) and BSA (figure 4(C)) constituted the inherent background and were subtracted. Figure 4B and 4C did not show water in BSA spectra.

·         Figure 4D and 4D should be presented with the same intensity scale to evaluate the improvement in the spectral variability when the water normalization procedure was used

·         The authors reported that after subtracting the inherent background from the raw spectra without water normalization (Figure 4(B)), the spectra showed negative features in the region 3200-3800 cm-1 Can the authors explain the origin of this negative bands

·         Spectrum of Beta carotene in Figure 3A should be comparable to the mean spectra showed in the figure 4A and that in figure 7A before BSA and water subtraction.

·         The authors should explain clearly the differences in the Raman intensity of these three carotenoids (figure 7B) with respect to the absorbance spectra presented in Figure 1

·         Can the authors explain why the Raman intensities of the three carotenoids increased after correction for self-absorption.

·         The scatter plot displayed in Figure 8(B) shows precise clustering of the spectra of each carotenoid compounds, and a clear differentiation between them. The derivative peak shape at 1519 is due to the shift in the intensity of this band between Lutein than in Beta Carotene. PCA was not necessary to identify the shifts between the 3 carotenoids. This shift is visible in the spectra of the figure 7C

·         The results of figure 8 and 9 could be obtained by simple comparison of Raman spectra of these three carotenoids or by pairwise subtraction Beta Carotene minus Zeaxanthin, Beta Carotene minus Lutein, and Lutein minus Zeaxanthin to identify the shifts and differences between these carotenoids.

·         The comparison of normalized spectra of Beta Carotene, Lutein and Zeaxanthin displayed in Figure 10 should be added to figure 7

Author Response

Please see the attachment. Responses are colour coded green

Round 2

Reviewer 1 Report

Accept in present form

Reviewer 2 Report

Major revisions were performed on this work as suggested by the reviewers. The results and discussion sections were changed with the respect of reviewers’ comments. This article was clearly improved and can be published in molecules